# Integrative High-Throughput RNAi Screening Identifies BRSK1, STK32C and STK40 as Novel Activators of YAP/TAZ

**DOI:** 10.3390/ijms26167810

**Published:** 2025-08-13

**Authors:** Mandeep K. Gill, Siyuan Song, Tania Christova, Liliana Attisano

**Affiliations:** Department of Biochemistry, Donnelly Centre, University of Toronto, Toronto, ON M5S 3E1, Canada; mk.gill@mail.utoronto.ca (M.K.G.); siyuan.song@utoronto.ca (S.S.); tania.christova@utoronto.ca (T.C.)

**Keywords:** YAP/TAZ, Hippo pathway, RNAi screening, AMPK family members, triple-negative breast cancer, BRSK1, STK32C, STK40

## Abstract

Disruption of the Hippo pathway leads to activation of the YAP/TAZ transcriptional program which promotes tumor initiation, progression and metastasis in diverse cancers. Aggressive triple-negative breast cancers (TNBC) lack an effective therapy; thus, inactivating YAP and TAZ has emerged as an attractive approach and a new treatment modality. Thus, we performed two complementary high-throughput RNAi-based kinome screens to uncover cancer-associated activators of YAP/TAZ in two TNBC cell lines, MDA-MB231 and MDA-MB468. Integrated analysis that combined a YAP/TAZ localization screen with a TEAD-luciferase reporter screen, identified novel regulators including BRSK1, STK32C and STK40. The AMPK family members NUAKs, MARKs and SIKs are known to inhibit the Hippo kinase cassette; here, we uncover BRSK1, another AMPK family member as a regulator of YAP/TAZ. We also reveal that two poorly studied kinases, STK32C, a member of the AGC family, and STK40, a pseudokinase, can also inhibit the activity of YAP/TAZ. Thus, our studies expand the repertoire of known AMPK family members and reveal two new kinases that modulate the Hippo pathway and may play a role in YAP/TAZ driven breast cancers. Further analysis of other screen hits may similarly uncover new regulators that could be targeted for therapeutic interventions.

## 1. Introduction

Breast cancer (BC) is the most common worldwide malignancy in females with an estimated occurrence of 2.3 million new cases and 670,000 deaths in 2022 [1]. Although there have been significant therapeutic advances, a number of clinical challenges remain, particularly with regard to the development of targeted therapies for BC lacking estrogen receptors, progesterone receptors and human epidermal growth factor receptor-2, also known as triple-negative breast cancers (TNBC) [2,3]. This form accounts for approximately 10–15% of all BC cases and is generally a more aggressive type with poor prognosis [4]. For TNBC patients, chemotherapy is still the primary treatment option, but many tumors acquire resistance, resulting rapid relapse or metastasis after adjuvant therapy [2]. Therefore, it is essential to identify new therapeutic targets for TNBC.

The Hippo signaling cascade is an evolutionarily conserved pathway that plays an important role in regulating cell proliferation, tissue homeostasis and organ size [5,6,7]. The transcriptional co-activators Yes-associated protein (YAP) and the related transcriptional coactivator with PDZ-binding motif (TAZ, also called WWTR1) are the major downstream effectors of the Hippo pathway. YAP and TAZ are negatively regulated by phosphorylation through the core Hippo components, mammalian sterile 20-like kinase (MST1/2), large tumor suppressor (LATS1/2) kinases, and the scaffold proteins Salvador homolog 1 (SAV1) and Mps One Binder kinase activator protein 1 (MOB1) [5,6,7]. In response to diverse internal and external stimuli, such as cell contact, polarity, metabolic and nutrient status or altered mechanotransduction, the Hippo pathway is activated, in which MST1/2, with the adaptor protein SAV1, phosphorylates and activates LATS1/2 as well as the LATS binding protein, MOB1. Activated LATS1/2 then phosphorylates YAP/TAZ resulting in cytoplasmic sequestration and subsequent degradation [5,6,7]. The dysregulation of the Hippo pathway leads to the nuclear accumulation of YAP/TAZ, where YAP/TAZ bind to transcription factors, predominantly TEADs (TEAD1–4), but also others, such as SMADs, RUNXs, p73, AP-1 and TBX5, and drives the expression of genes involved in cell proliferation, apoptosis and oncogenic transformation [5,6,8,9,10,11]. In addition, cross-talk of the Hippo pathway with other signaling pathways such as TGFβ, GPCRs, MAPK, Wnt, Notch, PI3K and Hedgehog can regulate the activity of YAP/TAZ [5,6,10,12].

There is growing evidence that indicates that YAP and TAZ are key players in promoting a broad range of cancer-associated properties such as proliferation, cell survival, migration, tumor initiation and metastasis in diverse tumors including BC [8,9,10,11,12,13,14,15,16]. For instance, TAZ expression is associated with TNBC diagnosis, BC tumor grade and unfavorable patient prognosis [13,15]. Overexpression of YAP/TAZ promote tumorigenic and metastatic processes including cell proliferation, migration, and epithelial–mesenchymal transition (EMT) in human immortalized mammary epithelial cells [13]. In addition, elevated TAZ levels in the nucleus induce cancer stem cell (CSC)-like activity and promote resistance in certain BC drug therapies [16]. Moreover, YAP/TAZ has been linked to BC-associated metastasis by inducing metabolic reprogramming or regulating the activity of hypoxia-inducible factor-1 [14]. Thus, YAP and TAZ play an important role in breast cancer initiation, progression and metastasis.

Restoring the activity of the Hippo kinase cassette or targeting YAP/TAZ represents two potential therapeutic approaches to treat YAP/TAZ driven cancers [8,17,18]. Indeed, compounds that inhibit the interaction of YAP/TAZ with TEADs have been developed, some of which entered Phase I clinical trials [19,20,21,22]. However, targeting the Hippo pathway upstream of YAP-TEAD is a major challenge given that the core kinase cassette lacks appropriate druggable targets, such as MST and LATS, which are tumor suppressive. Moreover, upstream signals that modulate the Hippo pathway are emerging, but still remain incompletely understood. Studies have shown that cell polarity complexes and the physical and mechanical properties of cells influence this pathway, but how these cues are sensed and transduced to Hippo pathway components requires further investigation. Identifying novel regulators could expand not only our understanding of signaling from the cell surface to the core components of the Hippo pathway, but also provide therapeutic targets for treating BC.

As YAP/TAZ localization and transcriptional activity are the key readouts of Hippo pathway activity, we applied a high-throughput RNAi screening approach involving a YAP/TAZ subcellular localization assay and a TEAD-reporter screen which measures YAP/TAZ transcriptional output, using two TNBC cell line models. By integrating the results of these screens, we expect to uncover new modulators of YAP/TAZ in a breast cancer-relevant context. We identified many known Hippo pathway components including AMPK family members such as NUAK2, MARK3 and MARK4, confirming fidelity of the screen. Moreover, we uncovered new regulators including the AMPK family member, BRSK1 and two poorly studied kinases, STK32C and STK40 as positive regulators of YAP/TAZ activity. Therefore, our findings emphasize the potential of high throughput screening approaches to identify novel YAP/TAZ regulators, and how this strategy can be further used to uncover new targets that may be appropriate for development into a cancer therapeutic.

## 2. Results

### 2.1. High-Throughput RNAi Screening

To identify novel regulators of the YAP/TAZ/Hippo pathway in BC, we undertook two complementary high-throughput RNAi screening approaches using a human kinome library designed to target 720 kinases or kinase regulatory subunits (Figure 1a). To identify TNBC cell lines with an inactive Hippo pathway appropriate for screening, we selected two aggressive TNBC cell lines, MDA-MB231 and MDA-MB468, for further analysis. Examination of YAP/TAZ subcellular localization by immunofluorescence microscopy revealed that in both cell lines, YAP/TAZ was predominantly localized in the nucleus (Figure 1b) indicative of an inactive Hippo pathway. We also assessed YAP/TAZ transcriptional activity using a TEAD-luciferase reporter in which YAP/TAZ associates with the TEAD transcription factors to induce expression of the luciferase reporter gene. Concordant with the nuclear localization of YAP/TAZ, both lines displayed high levels of TEAD-reporter activity (WT), which was markedly reduced by abrogating the expression of YAP using siRNAs. No luciferase activity or effects of siYAP were observed on a variant reporter harboring mutant TEAD binding sites (MUT) (Figure 1c). Thus, both lines display prominent YAP/TAZ activity and are appropriate for screens to identify Hippo pathway regulators.

### 2.2. YAP/TAZ Imaging Screen Identifies Regulators of YAP/TAZ Subcellular Localization

YAP/TAZ subcellular localization is an indicator of Hippo pathway activity, thus for the first screen we monitored the effects of siRNAs on YAP/TAZ localization using high-throughput automated immunofluorescence microscopy, optimized in a 384-well format to enhance throughput. Cells were seeded to ensure subconfluency at the assay endpoint, which maximizes YAP/TAZ nuclear localization and activity. Cells were reverse transfected with siRNAs (a pool of four individual siRNAs) targeting individual genes from the kinome library and after 48 h, cells were fixed, stained with anti-YAP/TAZ antibodies and DAPI to identify nuclei and subjected to automated image analysis. The distribution of the fluorescence signal between the cytoplasm (C) and the nucleus (N), referred to as the C/N ratio, normalized to the distribution in siCTL-transfected cells was quantitated and Z-scores determined. Both cell lines were screened in two independent biological replicates.

To assess screen performance, we first analyzed the effects of positive and negative (scrambled, nontargeting siRNA; siCTL) control. We previously identified MARK4 as an activator of YAP/TAZ in MDA-MB231, but not in MDA-MB468 cells, that binds and phosphorylates Hippo core kinases MST1/2 and SAV, to inhibit LATS1/2 activity [17]. Thus, we selected siMARK4 as our positive control. Consistent with expectations, in MDA-MB231 cells, loss of MARK4 strongly induced cytoplasmic localization of YAP/TAZ with a frequency distribution of positive Z-scores that did not overlap with that of the negative control (siCTL) (Figure 1d,e). In contrast, in MDA-MB468 cells, siMARK4 largely overlapped with the siCTL transfected cells (Figure 1d,e) as expected [17]. Next, we assessed the reproducibility of the biological replicate runs. Frequency histograms, showing the number of siRNAs and the degree of the effect on YAP/TAZ localization, showed overlapping distributions between the two independent runs in both TNBC cell lines (Figure 2a). The results for each biological replicate were independently rank-ordered by Z-score (Figure 2b), and the common genes within the top 25% for each run that promoted cytoplasmic localization of YAP/TAZ upon knockdown (i.e., genes that act as Hippo pathway inhibitors and/or YAP/TAZ activators) were examined first (Appendix A).

This identified 130 and 84 hits in MDA-MB231 and MDA-MB468 cell lines, respectively, with 38 hits being common to both cell lines (Figure 2c, Appendix A). Notably, the characterization of NUAK2 that emerged from the overlap of these two screens has been previously published [23]. Since YAP/TAZ is predominantly localized in the nucleus in these cell lines, identification of siRNAs that further enhance nuclear accumulation (i.e., Hippo pathway activators) is limited. Indeed, even when using a low Z-score cut-off of <−1.8, only five or three common hits were identified in MDA-MB231 and MDA-MB468 cells, respectively, none of which were common in the two lines (Figure 2c).

### 2.3. TEAD-Luciferase Reporter Screen Identifies Regulators of YAP/TAZ Activity

To complement the imaging screen, we conducted a second screen that directly monitored the effects of siRNAs on YAP/TAZ transcriptional activity by using the TEAD-reporter assay. To perform this screen, cells were plated in a 96-well dish, so that cells were subconfluent by the end of the assay, and 24 h after introduction of siRNAs, cells were transfected with the TEAD-luciferase reporter and a β-galactosidase expression vector. After another 24 h, TEAD-reporter activity was measured by luciferase assay normalized to β-galactosidase activity and normalized luciferase readings were converted to fold over plate median (Appendix A). As expected, loss of YAP, the positive control, inhibited TEAD-luciferase reporter activity in both TNBC cell lines (Figure 3a). The results for each cell line were independently rank-ordered by the fold change, and genes whose loss decreased TEAD-reporter activity (i.e., Hippo pathway inhibitors) were determined using a fold change cut-off < 0.5 (Figure 3b). This identified 160 and 102 hits in MDA-MB231 and MDA-MB468 cell lines, respectively (Appendix A), with 45 hits being common to both cell lines (Figure 3c). In contrast to the YAP/TAZ localization screen, the TEAD-reporter screen also identified many siRNAs that enhanced TEAD-reporter activity (i.e., genes acting as Hippo activators and/or YAP/TAZ-TEAD inhibitors). Specifically, using a fold change cut-off > 2 fold, 142 and 111 hits were identified in MDA-MB231 and MDA-MB468 cells, respectively, with 46 hits being common to both cell lines (Figure 3c, Appendix A). This increase in hits likely reflects the identification of general transcriptional regulators along with potential pathway-specific modulators of YAP/TAZ.

### 2.4. Integration of Screens Identifies Known and Novel Hippo Pathway Regulators

To increase confidence in bona fide Hippo pathway regulators, genes whose silencing promoted both cytoplasmic localization of YAP/TAZ and decreased TEAD-reporter activity were determined. This identified 41 and 18 hits in MDA-MB231 and MDA-MB468 cells, respectively, with 4 hits being common to both cell lines (Figure 4a–d). Only one gene whose silencing led to enhanced nuclear YAP/TAZ and increased TEAD-reporter activity was identified (in MDA-MB231 cells), consistent with expectations given the high basal level of nuclear YAP/TAZ (Figure 4a). Accordingly, we focused further efforts on Hippo pathway inhibitors. Analysis of those hits that were common to both cell lines revealed four genes, including NUAK2, EPHA3, PIP4K2B and PRKAB1 (Figure 4d), all of which have been directly confirmed as components acting in pathways known to modulate Hippo signaling. We previously showed that NUAK2 negatively regulates the Hippo cassette by phosphorylating and inactivating LATS1/2 [18] and that chemical inhibition of NUAK2 restores Hippo pathway activity, suppresses YAP/TAZ and inhibits the growth of cancer cell lines in vitro and in mammary tumors in mice [18], with others showing similar results in liver cancer [19]. The Ephrin receptor, EPHA3, functions in development and stem cell maintenance and can promote tumorigenesis in diverse cancers [20,21,22]. While this gene has not been directly linked to Hippo, the related EphA2 receptor was shown to regulate YAP in breast and gastric cancer cells [23,24] and other EPHA receptors (EPHA4, EPHA5, EPHA7 and EPHA8) emerged as Hippo pathway activators in an siRNA screen in HEK293T cells [25]. Another hit, PIP4K2B, plays a role in insulin resistance and promotes tumorigenesis by regulating PI3K signaling [26], a pathway which is known to inhibit Hippo [27,28,29,30]. The fourth hit, PRKAB1, is a regulatory subunit of AMPK, with AMPK having been previously demonstrated to promote Hippo pathway activity [31,32,33]. A few hits common to both cell lines, but detected in only one screen type, include NEK2 (in the localization assay), which was shown to interact directly with the Hippo pathway components SAV1 and MST1/2 to regulate centrosome disjunction [34] and NEK8 (in the TEAD-reporter assay), a ciliopathy gene that inhibits LATS-mediated phosphorylation of TAZ [35]. As individual cell lines may display differences in Hippo pathway regulation, we also assessed hits common to the localization and TEAD-reporter screens but appearing in a cell-line specific manner (Figure 4b,d). This revealed known regulators of YAP/TAZ such as ILK, CDK5, CDK8 and FGFR2 in MDA-MB231 cells [30,36,37,38] and MARK3 and CIT [39,40,41] in MDA-MB468 cells. Although not previously described, SPHK2, which phosphorylates sphingosine to produce S1P, a GPCR ligand known to inhibit the Hippo pathway, was also identified in MDA-MB231 cells [42]. Thus, both the individual and overlayed screens successfully identified a wide range of known Hippo pathway regulators that function through diverse mechanisms. This indicates that novel hits emerging from the screens may also be bona fide Hippo pathway regulators. Accordingly, we selected three novel hits, namely BRSK1, STK32C and STK10 for further validation as detailed below.

### 2.5. BRSK1, an AMPK Family Member Positively Regulates YAP/TAZ Activity

The AMPK family includes 14 members, several of which (AMPK, NUAK1 and 2, MARK1-4, and SIK2) have been shown to regulate Hippo pathway activity [17,18,19,25,31,32,33,39,40,43,44,45,46,47]. Thus, the emergence of BRSK1, another AMPK family member, as a hit in MDA-MB231 cells was of particular interest.

BRSK1, a ubiquitously expressed gene with the highest expression in the nervous system, is involved in cell cycle control, development of neuronal polarity [48,49,50], and in a few studies, it was found that it might be involved in tumorigenesis [51,52]. In our screen, we observed that the loss of BRSK1 resulted in cytoplasmic localization of YAP/TAZ and suppression of the TEAD-reporter in MDA-MB231 cells. We first verified the screen results and confirmed that the abrogation of BRSK1 reduced the nuclear accumulation of YAP/TAZ in MDA-MB231 cells (Figure 5a). Concordantly, the loss of BRSK1 expression inhibited the TEAD-luciferase reporter, but had no effect on a variant reporter harboring mutant TEAD binding sites (Figure 5b).

Next, we examined the effect of loss of BRSK1 expression on endogenous YAP/TAZ target gene expression. Real-time PCR analysis showed that loss of BRSK1, using either a pool or two single siRNAs, reduced the expression of YAP/TAZ target genes *ANKRD1, CTGF* and *CYR61* in MDA-MB231 cells (Figure 5c,d). BRSK1 was not identified as a hit in MDA-MB468 cells in our screen and consistent with this, no reduction in expression of YAP/TAZ target genes was observed in MDA-MB468 cells (Figure 5e). Nuclear YAP/TAZ drives breast cancer cell proliferation, and concordantly, siRNA mediated abrogation of BRSK1 expression suppressed the growth of MDA-MB231 cells (Figure 5f). Therefore, these findings suggest that BRSK1 is a positive regulator of YAP/TAZ and that loss of BRSK1 can attenuate the growth of breast cancer cells.

Thus, AMPK family members, either previously known or identified in our study as inhibitors of the Hippo pathway, include NUAK2, NUAK1, MARK2, MARK3, MARK4, SIK2 and BRSK1. We next used the cBio Cancer Genomics Portal (http://cbioportal.org) [53,54,55] to determine the expression levels of these genes in individual patient samples within TCGA PanCancer Atlas, Breast Invasive Carcinoma dataset [53,54,55]. Interestingly, analysis of the oncoprint revealed that each of these genes were overexpressed in different patient tumors with limited overlap (Figure 6). Notably, consistent with prior studies, both YAP and TAZ (WWTR1) were also overexpressed in a subset of samples [13,15,16] that was distinct from those with high expression of AMPK family members (Figure 6). These results indicate that in individual invasive breast cancer tumors, Hippo pathway inactivation can occur via diverse mechanisms including overexpression of diverse AMPK family members or YAP and TAZ and suggests a widespread role for YAP/TAZ activation in breast cancer tumorigenesis.

### 2.6. STK32C and STK40 Are Positive Regulators of YAP/TAZ

Among other regulators, our screen identified several other kinases not previously linked with the Hippo pathway that appeared as top-ranked hits, two of which, STK32C and STK40, were in MDA-MB231 cells. STK32C and STK40 are two poorly characterized kinases that are ubiquitously expressed, with the highest expression in the human brain.

STK32C, a member of the AGC family of kinases has been implicated in HMGB1 signaling in bladder cancer progression [56] and was recently shown to be involved in mediating resistance to doxorubicin in TNBC cells [57], while STK40, a pseudokinase, was shown to be involved in mesoderm development, migration of endothelial cells and, of relevance here, in promoting the growth of triple negative breast cancer cells [58,59,60]. In our screen, silencing of STK32C and STK40 resulted in cytoplasmic localization of YAP/TAZ and suppression of the TEAD reporter. We confirmed that siRNA-mediated abrogation of the expression of STK32C or STK40 reduced the nuclear accumulation of YAP/TAZ in MDA-MB231 cells (Figure 7a). Concordantly, the loss of STK32C or STK40 suppressed the expression of the endogenous YAP/TAZ target genes, *ANKRD1, CTGF* and *CYR61* (Figure 7b–e) and reduced the growth of MDA-MB231 cells (Figure 7f,g). Thus, this data suggests that STK32C and STK40 are positive regulators of YAP/TAZ, and their loss can attenuate growth of breast cancer.

## 3. Discussion

Worldwide, BC is the most frequent life-threatening cancer in women and is the leading cause of cancer deaths among women [1]. While considerable progress has been made in the development of various treatments, further efforts are required to establish targeted therapies that control tissue growth and homeostasis, as disruption of the pathway contributes to cancer development [5,6,8,9,10,11]. YAP/TAZ is overexpressed in most solid tumors, and enhanced nuclear accumulation of YAP/TAZ is associated with poor prognosis in a wide range of cancers, including BC [8,9,10,11,13,14,15,16]. Several upstream signals derived from apical-basal polarity complexes, physical-mechanical signals, energy stress, GPCR signaling and others can activate the Hippo pathway which results in phosphorylation of YAP/TAZ by LATS1/2 and thereby cytoplasmic retention and subsequent degradation [5,6,8,9,10,11]. However, much remains to be understood on how these signals are transduced to core components. Here, we performed high-throughput RNAi screening with the goal of identifying novel regulators of the YAP/TAZ-Hippo pathway that may expand our understanding of how cues from the cell surface are transduced to the core Hippo components or may modify YAP/TAZ activity.

In the human genome, there are approximately 538 kinase genes, and these kinases play a central role in intracellular signal transduction and complex cellular functions such as cell proliferation and death. Aberrant expression or activation of kinases may promote tumorigenesis, malignancy and drug resistance. Numerous oncogenic kinases have been discovered in different types of cancer and for a subset, highly selective inhibitors have been developed as targeted therapeutics [24]. Unfortunately, targeting Hippo core kinase components MST1/2 or LATS1/2 is not a viable option as this would promote tumor development. However, the phosphorylation status of Hippo core kinases and of YAP/TAZ determines their activity, and given that kinases are conventional drug targets, we thus elected to focus on the kinome in our screens. While other groups have performed screens to identify novel Hippo pathway regulators [25,26,27,28,29,30] here, we used triple-negative breast cancer cells to enhance the likelihood of identifying breast cancer relevant regulators. We undertook two complementary high-throughput RNAi screening approaches by combining a YAP/TAZ localization screen with a TEAD-luciferase reporter screen. This screening procedure successfully identified many known regulators of YAP/TAZ including ILK, CDK5, CDK8, FGFR2, CIT, NEK2 and NEK8 [31,32,33,34,35,36,37,38,39,40]. Moreover, our integrated analysis identified several kinases not previously linked with the Hippo pathway that regulated YAP/TAZ subcellular localization and activity. As for all knockdown/knockout approaches, a limitation of this and all siRNA screens is that functional redundancy can result in false negatives. For instance, individual knockdown of MST1 or MST2 failed to induce YAP/TAZ activation. Other limitations of RNA interference approaches include issues with efficiency of silencing and the possibility of off-target effects, though these are partially mitigated by the use of siRNA pools. Nevertheless, findings from our integrated screen support previous knowledge and rationalize further analysis to uncover new drug targets for cancer treatment.

Amongst the identified potential novel activators of YAP/TAZ, we further studied two poorly characterized kinases that appeared as top-ranked hits, STK32C and STK40. The loss of expression of these kinases promoted cytoplasmic localization and decreased the transcriptional activity of YAP/TAZ, resulting in a concomitant decrease in cell growth. This data suggests that STK32C and STK40 enhance YAP and TAZ transcriptional activity to promote cancer cell growth. Two recent studies have indicated that both of these kinases are overexpressed in TNBC and in the case of STK32C, that this is associated with unfavorable prognosis in doxorubicin-treated TNBC patients [41,42]. Our findings suggest that the tumor-promoting effects of STK32C and STK40 in TNBC may be achieved through the regulation of YAP/TAZ. Further studies are required to define exactly how STK32C and STK40 modulate the Hippo-YAP/TAZ pathway.

Our screens also identified several both known and novel members of the AMPK-related family. The AMPK family includes the prototypic AMPK, which is responsible for maintaining cellular energy, and thirteen additional members, whose catalytic domains share sequence homology with AMPK, including NUAK1, NUAK2, MARK1, MARK2, MARK3, MARK4, SIK1, SIK2, SIK3, BRSK1, BRSK2, MELK and SNRK [43,44]. Several members of the AMPK-related family can regulate Hippo pathway activity, and a subset of these, including the AMPK regulatory subunit, PRKAB1, as well as NUAK2, MARK3 and MARK4 emerged in our screens. Activation of the prototypic AMPK in response to glucose deprivation leads to inhibition of YAP activity by either directly phosphorylating YAP to disrupt the YAP-TEAD interaction, or by phosphorylating AMOTL1 that promotes LATS activity [45,46,47]. The specific role of PRKAB1, an AMPK regulatory subunit, has not been examined, but as loss of PRKAB1 inhibited nuclear YAP/TAZ localization in our screen, this suggests that PRKAB1 may generally act to inhibit AMPK activity. NUAK2, a frequently amplified gene in tumors including breast cancer, and the related NUAK1, have been shown to inactivate the Hippo kinase cassette to promote tumorigenesis in breast, liver (NUAK2) and colorectal (NUAK1) cancers, as well as in driving fibrosis (NUAK1) in kidney and other tissues in a YAP-dependent manner [23,48,49,50]. Our identification of NUAK2 but not NUAK1 in the TNBC lines, an observation we confirmed in separate studies [23], suggests that NUAK2, rather than NUAK1, may play a more prominent role in promoting YAP/TAZ activity in breast cancer. Members of the MARK subfamily, including MARK2, 3 and 4, regulate the Hippo kinase cassette in diverse models including Drosophila, mice and human cancer cell lines by phosphorylating MST1/2, SAV [29,51,52,53,54,55] and, as found in a recent study, NF2 and YAP/TAZ [53]. In our screen, we identified MARK4 in MDA-MB231 cells and MARK3 in MDA-MB468. Interestingly, a recent study reported that concomitant inhibition of both MARK2 and MARK3 is required to prevent YAP/TAZ function in diverse carcinoma and sarcoma contexts, highlighting the redundant nature of MARK family members. In Drosophila, similar to NUAKs and MARKs, SIKs were shown to activate Yorkie (YAP/TAZ ortholog) by phosphorylating the Hippo core component, Sav [30]. We did not identify any of the three SIKs in our screen possibly due to redundancy or cancer context. The remaining AMPK-related family members, BRSK1, BRSK2, SNRK and MELK have not been linked to the Hippo pathway, but of these, BRSK1 (but not BRSK2) was identified as a hit in our screen. BRSK1 and the related BRSK2 are poorly studied kinases known to play a pivotal role in cell cycle control and development of neuronal polarity [56,57,58]. We demonstrated that loss of BRSK1 promoted cytoplasmic localization of YAP/TAZ, diminished the transcriptional activity of YAP/TAZ and attenuated cell growth. These findings suggest that BRSK1, similar to NUAKs and MARKs, is a positive regulator of YAP/TAZ. Interestingly, analysis of a breast cancer patient cohort at the level of individual patient samples revealed that each AMPK family member was overexpressed in different individual tumors with limited overlap between the patient tumors (Figure 7). Thus, excessive YAP/TAZ activity can occur through Hippo pathway inactivation via distinct AMPK family members. We also observed that different subsets of patient tumors overexpressed either YAP or TAZ (WWTR1). Thus, while the contribution of each AMPK family kinase and/or YAP and TAZ is relatively small, when considered in combination, it is indicative of a key role for YAP/TAZ activation in breast cancer tumorigenesis. Analysis of the presence of changes in these components in patients also has the potential to provide individualized therapeutic regimens, based on the presence of distinct alterations, such as treatment with NUAK inhibitors for patients overexpressing NUAKs or YAP/TAZ-TEAD inhibitors for patients with amplifications of YAP or TAZ. Although AMPK-related family members promote the pro-oncogenic activity of YAP/TAZ, the molecular mechanisms of this action appear to be different, such as NUAKs inhibiting LATS1/2 [23,48,49,50] and MARKs targeting MST1/2, SAV and NF2 [29,51,52,53,54,55]. Thus, additional studies are required to fully appreciate the extent of conservation and overlap of the biological functions and molecular mechanisms for all AMPK family members.

## 4. Materials and Methods

### 4.1. Cell Culture

MDA-MB231 and MDA-MB468 were cultured in Roswell Park Memorial Institute (RPMI) 1640 medium (Gibco, 11875093, Waltham, MA, USA) and Dulbecco’s modified Eagle’s medium: nutrient mixture F-12 (DMEM:F-12) (Gibco 11320033) (1:1) containing 5 and 10% fetal bovine serum (FBS) (Gibco, 12483-20), respectively. Cell lines were obtained from American Type Cell Collection (ATCC, Manassas, VA, USA) and monitored for mycoplasma contamination using the MycoAlert Mycoplasma Detection Kit (Lonza, LT07-703, Basel, Switzerland). MDA-MB231 cell line was authenticated by short tandem repeat DNA profiling.

### 4.2. High-Throughput RNAi Imaging Screen

The high-throughput (HTP) RNAi screen was performed at the SMART Robotics Facility, Lunenfeld Tanenbaum Research Institute, Toronto, ON, Canada (http://nbcc.lunenfeld.ca, accessed on 8 August 2025). For the HTP imaging screen, cells were plated in 384-well plates (at 0.6 × 10^3^ cells/well for MDA-MB231 and 1.5 × 10^3^ cells/well for MDA-MB468). Cells were reverse transfected with 40 nM siRNAs derived from a siGENOME SMARTpool siRNA Human Kinome Library (Dharmacon, G-003505, Cambridge, UK) using Lipofectamine RNAiMAX (Invitrogen, 13778075, Life Technologies, Carlsbad, CA, USA). For controls, 72 wells were treated with Dharmacon non-targeting siRNA number 3 (siCTL) as negative controls and 16 wells with siMARK4 as positive controls. After 48 h cells were fixed with 4% paraformaldehyde solution, permeabilized with 0.5% Triton X-100/phosphate-buffered saline (PBS) and blocked with 2% bovine serum albumin (BSA)/PBS. Next, the primary YAP antibody (YAP, sc-101199, Santa Cruz, Dallas, TX, USA) was added at a 1:300 final dilution with 2% BSA/PBS. The antibody was incubated at 4 °C overnight. The following day, the secondary antibody (Alexa Fluor 488 goat anti-rabbit, A11305, Life Technologies) was added at a final 1:1000 dilution with 2% BSA/PBS containing 4′,6-diamidino-2-phenylindole dihydrochloride (DAPI) (D9542, Sigma-Aldrich, St. Louis, MO, USA) diluted 1:10,000 and incubated for 2 h at room temperature. After each step, wells were washed three times with 0.1% Tween-100/PBS. For manual siRNA-mediated gene knockdown, cells were reverse-transfected using Lipofectamine RNAiMAX (Invitrogen, 13778075, Life Technologies) according to the manufacturers’ instructions. The siRNA oligos are listed in Table 1.

### 4.3. High-Throughput RNAi TEAD-Reporter Screen

The TEAD-luciferase reporter wild-type construct, kindly provided by A. Gregorieff and J. Wrana (Lunenfeld Tanenbaum Research Institute, Mount Sinai Hospital, Toronto, ON, Canada) comprises 10 tandem multimerized TEAD binding sites (TGGAATGT), subcloned into a pGL3-Basic firefly luciferase reporter (Promega, E1751, Madison, WI, USA), fused to a minimal Hsp70 promoter as described previously [59]. Cells were plated in 96-well plates (at 0.9 × 10^4^ cells/well for MDA-MB231 and 3 × 10^4^ cells/well for MDA-MB468. Cells were first reverse transfected with siRNAs with 8 wells in each plate treated with Dharmacon non-targeting siRNA number 3 (siCTL) as negative controls and 4 wells of siYAP as positive controls. After 24 h cells were forward transfected with the TEAD-luciferase reporter along with a β-galactosidase expression vector using Lipofectamine LTX (Invitrogen, A12621, Life Technologies, Carlsbad, CA, USA) according to the manufacturer’s instructions. At 24 h post-cDNA transfection, cells were lysed in lysis buffer (25 mM Tris, 2 mM DTT, 2 mM DCTA, 10% glycerol, 1% Triton X-100) (Sigma-Aldrich, T8787, St. Louis, MO, USA) and luciferase and β-gal assays were performed. Luciferase activities were normalized to β-galactosidase activity, and normalized luciferase readings were converted to fold over plate median. The manual TEAD-reporter assay was performed as described previously [23].

### 4.4. High-Throughput Image Analysis

Images were acquired on IN Cell Analyzer 6000 (GE Healthcare, Chicago, IL, USA), equipped with sCMOS camera (2048 × 2048) and 20X/045 Plan Fluor objective (Nikon, Mississauga, ON, Canada) with Corrective Collar 0–2.0. Images were captured in DAPI and FITC channels using the 2-D Widefield Image Mode and no binning. The images were saved as 16-bit TIFF per channel. Image analysis was performed with Columbus™ Image Data Storage and Analysis system version 2.3 (PerkinElmer, Waltham, MA, USA). Nuclei were initially segmented in the DAPI channel, followed by the cytoplasm in the FITC channel. Cells were categorized into two sub-populations based on the ratio between the FITC intensity in the cytoplasm mask and the FITC intensity in the nuclear mask. The percentage of cells with nuclear translocation (cells with a FITC ratio 0.4) or cytoplasmic translocation (cells with a FITC ratio 0.65) per well were calculated. To identify hits, the Z-score was determined in two independent runs.

### 4.5. RNA Extraction, Real-Time PCR and Gene Expression Analysis

Total RNA was purified from cultured cells using PureLink RNA Mini Kit (Life Technologies, 12183025), and cDNA was synthesized using oligo(dT) primers and M-MLV Reverse Transcriptase (Invitrogen, 28025-013). Real-time PCR was performed using SYBR Green master mix (Applied Biosciences, Salt Lake City, UT, USA, 4309155A) on the ABI Prism 7900 HT apparatus (Applied Biosystems, Foster City, CA, USA). Relative gene expression was quantified by ΔΔCt method and normalized to HPRT1 as previously described [23]. Statistical significance for pairwise data was determined using Student’s unpaired two-sided *t*-test. Graphical representation of the data was produced in Microsoft Excel version 16 for Mac or GraphPad PRISM 8 (GraphPad Software Inc., La Jolla, CA, USA). The primer sequences are listed in Table 2. An oncoprint depicting genes overexpressed in breast cancer patient samples from the TCAG dataset [60] was determined using cBioportal [61,62].

### 4.6. Immunofluorescence Microscopy

For immunofluorescence microscopy, cells were fixed with 4% paraformaldehyde, permeabilized with 0.5% Triton X-100 ( Sigma-Aldrich, T8787, St. Louis, MO, USA)/PBS for 10 min at each step and blocked with 2% BSA/PBS prior to the addition of primary antibody. To quantify YAP/TAZ localization, a minimum of 30 cells per experimental condition were counted, and data was plotted as the percentage of cells with equivalent nuclear and cytoplasmic (N=C), predominantly nuclear (N>C) or predominantly cytoplasmic (C>N) YAP/TAZ localization. Primary antibodies used were YAP1 (sc-101199, Santa Cruz, 1:300), and secondary antibodies were goat anti-mouse Alexa Fluor 546 (Invitrogen, A11029, 1:1000). Samples were counterstained with 4′,6-Diamidino-2-phenylindole dihydrochloride (DAPI) (D9542, Sigma-Aldrich).

## Figures and Tables

**Figure 1 ijms-26-07810-f001:**
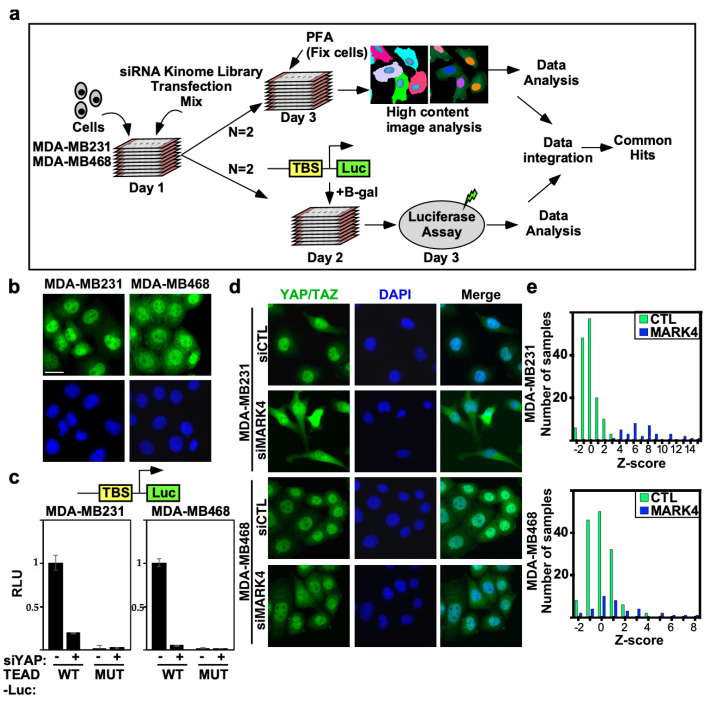
Analysis of high-throughput RNAi kinome screen controls and selected cell lines. (**a**) A schematic depicting the siRNA kinome screen. (**b**) Predominantly nuclear localization of YAP/TAZ in selected TNBC cell lines. Representative images from MDA-MB231 and MDA-MB468 cells showing localization of endogenous YAP/TAZ (green) with nuclei co-stained with DAPI (blue) were obtained by immunofluorescence confocal microscopy. Scale bar, 25 µm. (**c**) Loss of YAP suppresses wild-type (WT) but not mutant (MUT) TEAD-luciferase reporter activity in MDA-MB231 and MDA-MB468 cells. (siControl:−, siYAP:+). Data is plotted as the mean +/− SD (n = 3). (**d**) Loss of MARK4 promotes cytoplasmic localization of YAP/TAZ. Representative images from the screen collected by the IN Cell Analyzer 6000 (GE Healthcare, Chicago, IL, USA) are shown. Scale bar, 25 µm (**e**) The frequency distribution of controls combined from two independent runs is depicted as a histogram, where the number of samples out of a total of 144 siCtl and 36 siMARK4 at the given Z-score (determined on the basis of the entire screen) in each bin is plotted. Controls and siMARK4 samples are marked with green and blue bars, respectively.

**Figure 2 ijms-26-07810-f002:**
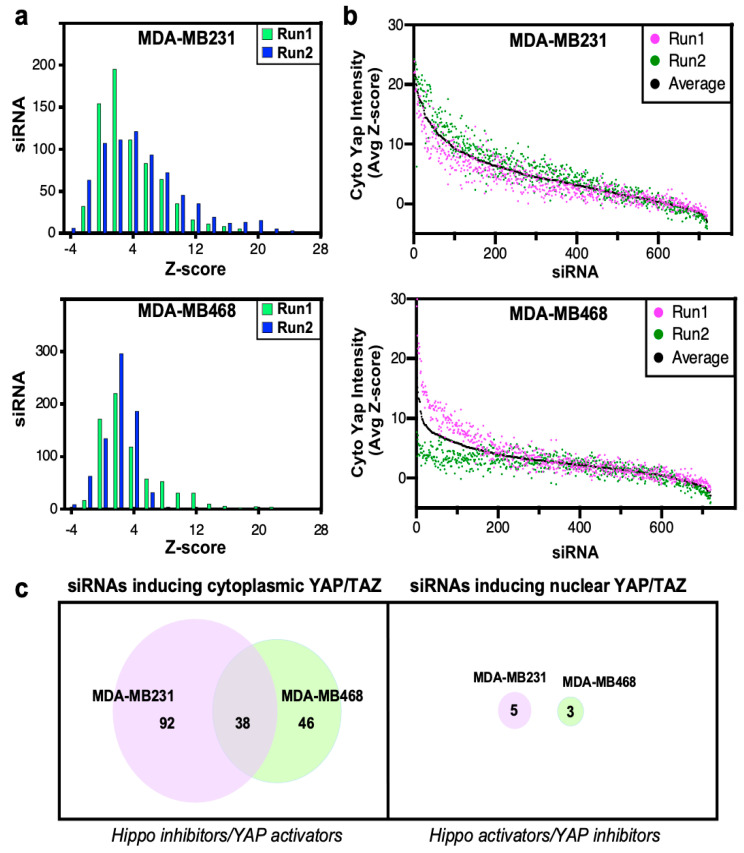
Analysis of high-throughput YAP/TAZ localization screen. (**a**) The frequency distribution of siRNAs is depicted as a histogram, where the number of siRNAs in each bin is plotted. (**b**) Screen results are rank-ordered by the average Z-score. (**c**) Venn diagrams indicating the number of candidate hits defined by the localization screens in the two cell lines.

**Figure 3 ijms-26-07810-f003:**
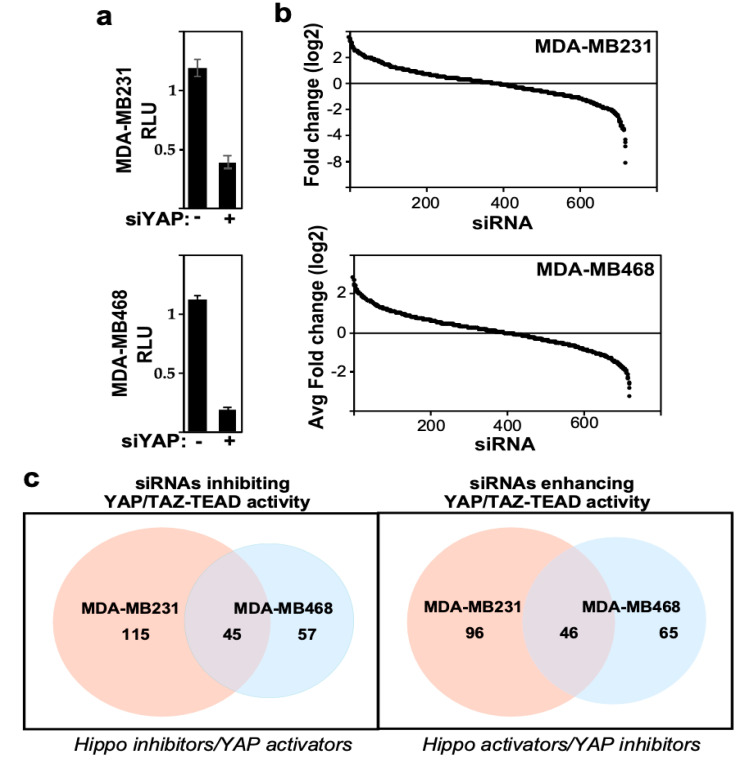
Analysis of high-throughput TEAD-reporter screen. (**a**) Loss of YAP reduces TEAD-reporter activity. Data is plotted as the mean ± SEM. (**b**) Screen results are rank-ordered by fold change. (**c**) Venn diagrams indicating the number of candidate hits defined by the TEAD-reporter screens in the two cell lines.

**Figure 4 ijms-26-07810-f004:**
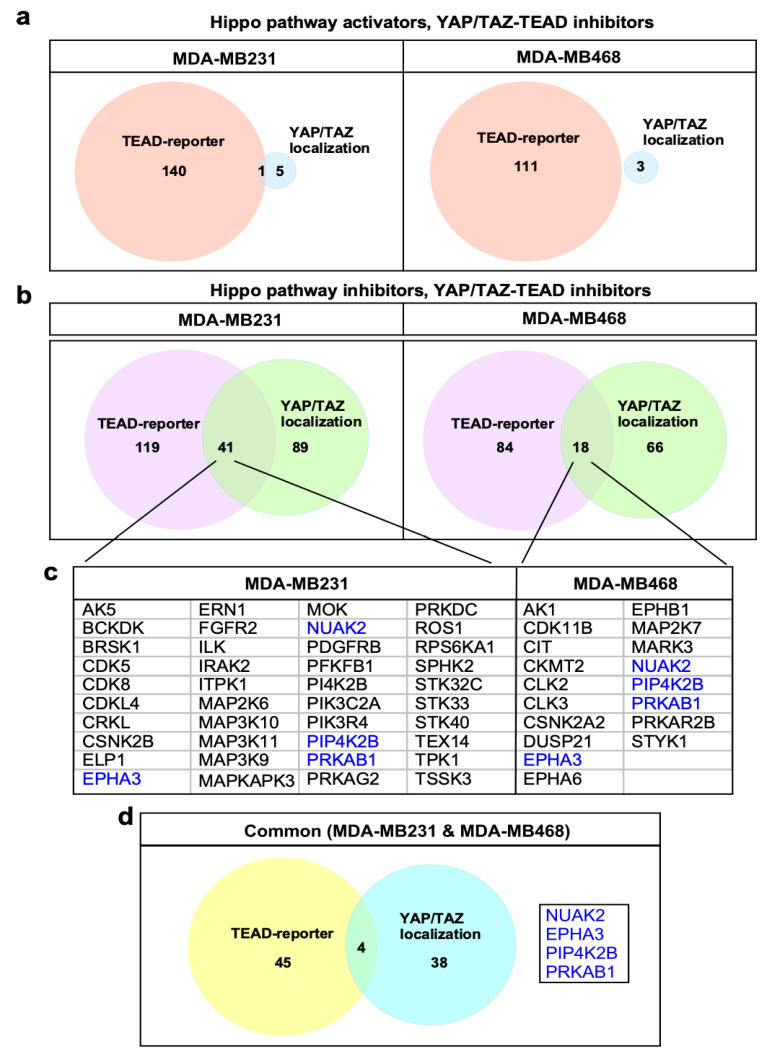
Identification of screen hits. (**a**,**b**) Venn diagram indicating the number of candidate hits defined by the two parallel screens. (**c**) Identified hits common to both localization and TEAD-reporter screens for each line are listed. The four common genes are highlighted in blue text. (**d**) Identified hits common to both localization and TEAD-reporter screens in both cell lines with 4 common genes listed.

**Figure 5 ijms-26-07810-f005:**
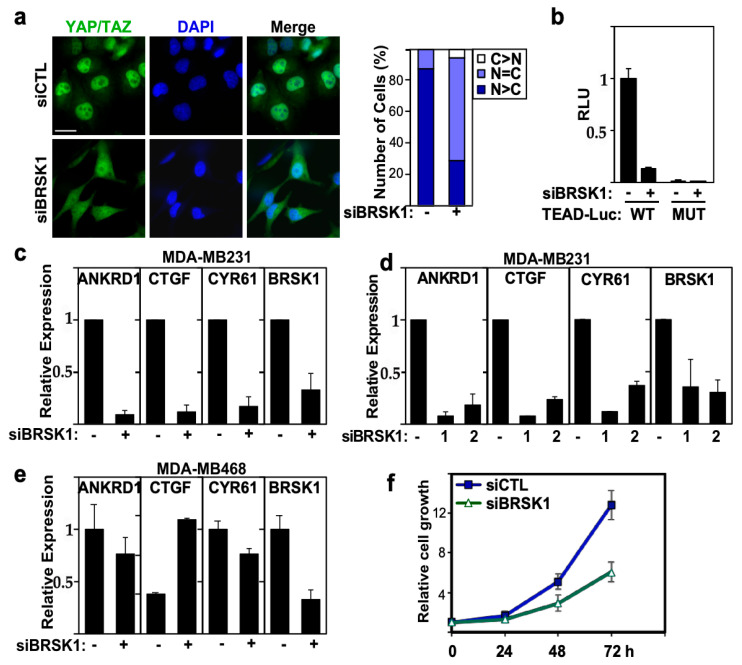
BRSK1 regulates YAP/TAZ localization, transcriptional activity and growth in MDA-MB231 cells. (**a**) Loss of BRSK1 expression promotes cytoplasmic localization of YAP/TAZ. Representative immunofluorescence image of endogenous YAP/TAZ (green) with nuclei co-stained with DAPI (blue) visualized by confocal microscopy in MDA-MB231 cells, transfected with control (−) or a pool of siRNAs targeting BRSK1 (left). The percentage of cells with equivalent nuclear and cytoplasmic (N + C), predominantly nuclear (N>C) or predominantly cytoplasmic (C>N) YAP/TAZ localization is plotted (right). Scale bar, 25 µm. (**b**) Loss of BRSK1 expression decreases TEAD-luciferase reporter activity. YAP/TAZ transcriptional activity was assessed using a wild-type (WT) and mutant (MUT) TEAD-luciferase reporter in MDA-MB231 cells, transfected with control (−) or a pool of siRNAs targeting BRSK1. Data is plotted relative to siCTL, and the mean ± SD of three independent experiments is shown. (**c**–**e**) Loss of BRSK1 expression decreases YAP/TAZ target gene expression. Relative expression of *ANKRD1, CTGF, CYR61* and knockdown efficiency of BRSK1, transfected as a pool or as individual (numbered 1 and 2) siRNAs, as measured by qPCR in MDA-MB231 (**c**,**d**) or MDA-MB468 cells (**e**) is plotted. The mean ± SD of three independent experiments is shown. (**f**) Loss of BRSK1 inhibits the growth of MDA-MB231 cells as measured by DAPI staining. Data is plotted as the mean ± SD of a representative experiment, n = 3.

**Figure 6 ijms-26-07810-f006:**
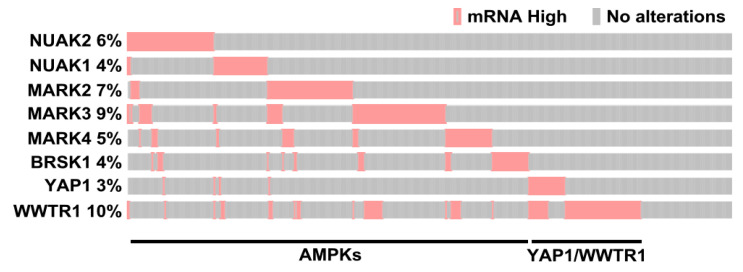
Overexpression of AMPK-related family members in breast cancer patient samples. An oncoprint generated in cBioportal depicts individual samples displaying overexpression (>2 standard deviations above the mean compared to diploids) of the indicated AMPK-related family members, known or identified in this study to act as YAP/TAZ activators and for YAP and TAZ. The plot was cropped to remove unaltered samples.

**Figure 7 ijms-26-07810-f007:**
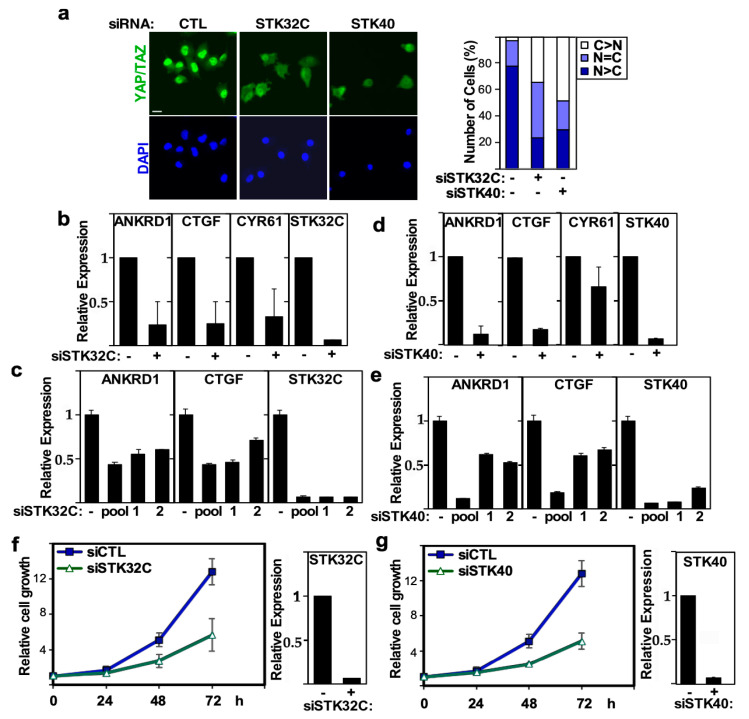
STK32C and STK40 regulates YAP/TAZ localization, transcriptional activity and growth in MDA-MB231. (**a**) Loss of STK32C or STK40 expression promotes cytoplasmic localization of YAP/TAZ. Representative immunofluorescence image of YAP/TAZ (green) with nuclei co-stained with DAPI (blue) in MDA-MB231 cells, transfected with control (−) or a pool of siRNAs, targeting STK32C or STK40 (left). The percentage of cells with equivalent nuclear and cytoplasmic (N + C), predominantly nuclear (N>C) or predominantly cytoplasmic (C>N) YAP/TAZ localization is plotted (right). Scale bar, 25 µm. (**b**,**c**) Loss of STK32C or (**d**,**e**) STK40 expression in MDA-MB231 cells, transfected with control (−), pool or individual (numbered 1 or 2) siRNAs decreases YAP/TAZ target gene expression. Relative expression of *ANKRD1, CTGF, CYR61* and knockdown efficiency of STK32C or STK40 as measured by qPCR is plotted. The mean ± SD of two independent experiments is shown. (**f**,**g**) Loss of STK32C or STK40 using siRNA pools (+) inhibits the growth of MDA-MB231 cells as measured by DAPI staining. Data is plotted as the mean ± SD of a representative experiment, n = 3. STK32C or STK40 knockdown efficiency (right) is plotted as mean ± range of a representative experiment.

**Table 1 ijms-26-07810-t001:** List of siRNAs.

Target Gene	Dharmacon Catalog Number
Non-targeting	D-001810-03
*BRSK1*	D-004619-05,06
*MARK3*	D-003517-02,03,04,10
*MARK4*	D-005345-01,02,05
*YAP*	D-012200-01,02,03,04
*STK40*	MQ-005348
*STK32C*	MQ-004615

**Table 2 ijms-26-07810-t002:** Sequences of primers for qPCR.

Target	Forward	Reverse
*HPRT1*	ATGGACAGGACTGAACGTCTTGCT	TTGAGCACACAGAGGGCTACAATG
*BRSK1*	CACGACGTCTACGAGAACAAGA	CAGGTAGTCGAATAGCTCACCC
*MARK3*	CTGGTGGAATGACACGACGA	AGGAATAGTGCTGTTTTCTTTGCC
*ANKRD1*	AGTAGAGGAACTGGTCACTGG	TGGGCTAGAAGTGTCTTCAGAT
*CTGF*	AGGAGTGGGTGTGTGGACGA	CCAGGCAGTTGGCTCTAATC
*CYR61*	CGAGGTGGAGTTGACGAGAAA	CTTTGAGCACTGGGACCATGA
*STK32C*	ACCGTGAGCGTCCAGTATG	CAGTGCAGACGGCCTTTGT
*STK40*	GAGAGCATCAGACAGAGGAG	ATGAATGGTCCAGCTCTCTT
*MARK2*	CACATTGGAAACTACCGGCTC	GGAGGAGTTCAGTTGAGTCTTGT

## Data Availability

The data presented in this study are available in the article and Appendix A.

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
