# Peer review of "Integrative High-Throughput RNAi Screening Identifies BRSK1, STK32C and STK40 as Novel Activators of YAP/TAZ"

_ijms, 2025, doi:10.3390/ijms26167810_

Round 1
Reviewer 1 Report
Comments and Suggestions for Authors
The authors report a kinome-wide RNAi screen to assess nuclear vs cytosolic YAP and TEAD activation using two different assay approaches; immunofluorescence high content imaging and a reporter assay. The authors have previously published a study with an identical RNAi screen using both two same model cell lines and the same high content assay with immunofluorescence staining of YAP to assess nuclear/cytosolic YAP (Gill MK et al., A feed forward loop enforces YAP/TAZ signaling during tumorigenesis. Nat Commun. 2018 Aug 29;9(1):3510. doi: 10.1038/s41467-018-05939-2. PMID: 30158528). Thus it seems the current report is a re-analysis of this same screen complemented with an additional TEAD reporter screen? However, the 2018 study by the authors doesn´t disclose the RNAi screen results in full so it is difficult to judge if this is the case or not. Either way, the authors should explain if this is the case or how the high content screen reported here is different. Second, related to the TEAD reporter assay used for the second screen, the authors should describe this reporter in more detail to allow evaluation of its utilization in the study and the identified “hits”, or at least give details where it was acquired since the details of the construct are difficult to find tracing back through the cited paper(s) (and the paper these cite; Serrano I et al., Inactivation of the Hippo tumour suppressor pathway by integrin-linked kinase. Nat Commun. 2013;4:2976. doi: 10.1038/ncomms3976. PMID: 24356468).
Main scientific notion following review of the manuscript affects the manuscript as a whole, as the assumption that the assay would only assess Hippo pathway related genes/proteins (as mentioned throughout the text) might not be the case. Following the recent surge of novel small molecule TEAD inhibitors that have been described an entered clinical development, it is becoming increasingly evident that YAP is regulated in response to several upstream pathways which may or may not involve the core hippo pathway and thus calling the study and the nominated hits as e.g., “Hippo pathway inhibitors” is somewhat misleading, especially without molecular studies explaining how the nominated hits are directly related to the Hippo pathway. The authors should thus revert to calling the assay and the identified hits simply as regulators of YAP/TAZ, according to the manuscript title. Last, as minimum level of validation of the reported findings, the authors should validate the knockdown efficacy of the siRNAs used in the pools for the key targets BRSK1, STK32C and STK40. If no functional antibodies are available for protein level validation of the knockdown efficacy, the authors could also use RNA seq in response to the siRNAs which would then allow also interesting assessments of effects on the genes on the broader YAP/TAZ signatures (e.g., Kanai R et al., Identification of a Gene Signature That Predicts Dependence upon YAP/TAZ-TEAD. Cancers (Basel). 2024;16(5):852. doi: 10.3390/cancers16050852. PMID: 38473214).
Detailed review comments:
Title: RNAi not mRNAi
Row 70-80: Latest approaches modulate downstream activity through targeting TEADs. One sentence about these approaches could fill this knowledge gap in the intro.
Row 110-123: Quite heavy in describing the methods used here. Trim text to focus on results and move method specific text to the appropriate section.
Row 141: “(b-c) Selection of cell lines” unnecessary? These sections are subsequently explained more in-depth.
Row 148: Same as above
Row 146: “YAP/TAZ transcriptional activity was assessed using a wild type (WT) and mutant (MUT) TEAD-146 luciferase reporter in cells with siControl (-) or siYAP (+).” Method description, unnecessary for figure text? (Compare with subsequent figure texts)
Row 157: What’s the reasoning behind using this specific z-score cutoff?
Row 167-174: Quite heavy in describing the methods used here. Trim text to focus on results and move method specific text to the appropriate section.
Row 289: Typo “Overexpressio” à Overexpression
Row 300-313: Unnecessarily busy and difficult to understand figure text. References to figures jump around (a-g à a à b-e à b,c ...) making it difficult to follow.
Fig 1E, could plot RFu (intensity) as % change signal to give reader a more concrete idea of the robustness. Is the z-score calculated against the mean and Stdv of only the negative controls?
Fig 2A, could replace with a scatter plot of the replicate experiments which would allow visual assessment of effect distribution and replicate correlation
Fig 2B, could replace/indicate for y-axis title cytosolic YAP intensity (z-score)
Fig 2C, since YAP is not only regulated downstream of Hippo pathway but in fact through several signaling pathways, instead of calling the “hits” hippo pathway inhibitors or activators authors could simply call the classes e.g., YAP cytosolic induction and YAP nuclear induction according to the RNAi effect
Fig 3C, similarly as Fig 2C, maybe instead of calling the “hits” hippo pathway inhibitors or activators the authors could call the classes TEAD activity inhibitors and TEAD activity inducers according to the RNAi effect
Overall: Figure formatting could be improved by using e.g. color coding in bar plots. Venn diagrams also need improving: If there’s no overlap in hits, there should be no overlap in the diagrams. Size of the venn circles could also represent the number of hits to make the figures clearer.
Reviewer 2 Report
Comments and Suggestions for Authors
This article describes a thorough high-throughput RNA interference method for identifying new Hippo pathway regulators in aggressive triple-negative breast cancer (TNBC) cells by activating the transcriptional cofactors YAP/TAZ. The authors discovered BRSK1, STK32C, and STK40 as novel positive regulators of YAP/TAZ using two parallel high-throughput RNAi screens: one based on the subcellular localisation of YAP/TAZ, and the other on TEAD-luciferase reporter activity. BRSK1, a member of the AMPK family, as well as the little-studied kinases STK32C and STK40, appear to promote YAP/TAZ activity and cancer cell growth. These findings highlight new potential therapeutic targets for TNBC through inhibition of the aforementioned molecules.
The article is particularly important in my opinion because it addresses a critical therapeutic problem – the lack of targeted therapies for TNBC. The association of YAP/TAZ overactivity with carcinogenesis, metastasis and resistance to therapy has been recognized, however the molecular mechanisms underlying it are not well understood. Through two combinatorial high-throughput experimental strategies, the article offers a new list of Hippo pathway regulators that deserve further study for the development of drug interventions. The proposals for BRSK1, STK32C and STK40 pave the way for new target molecules in cancers dependent on YAP/TAZ signaling.
In my opinion, the article needs some significant revisions.
- The abstract provides useful information, however it could benefit from a more focused formulation of the key finding. The use of technical details regarding the AMPK family and its subcategorization could be limited to one sentence, in order to give more space to explaining the importance of BRSK1, STK32C and STK40 as novel therapeutic targets.
- Although the Hippo pathway and its biological significance are clearly explained in the introduction, clarity in connecting the theoretical context and research strategy is missing. Including a brief description of what the new aspects of the localisation and TEAD-reporter screen combination are would make the reader more aware of the approach and the reasons behind the selection of the TNBC cell lines.
- The biological and clinical significance of some of the sections of the results is not described sufficiently (such as the OncoPrint data). The addition of a mention of how these results could lead to individualized therapeutic regimens and the intriguing observation that different patients overexpress different members of the AMPK family would be valuable.
- It is mentioned during the discussion that false negatives may be due to potential redundancy of function among kinases. Criticisms of the RNA interference approach, such as off-target effects or issues with the efficiency of silencing of individual genes, are not mentioned. A short paragraph acknowledging these criticisms would lend more credibility to the methodological discussion.
- Some grammatical errors are present, such as failure to place a comma after "kinases" in the phrase "STK32C, a member of AGC family of kinases has been implicated…." The writing is less readable in certain areas due to the absence of articles or joining very long sentences without punctuation. The presentation could be much improved with a linguistic clarity edit.
Some minor issues with grammar and phrasing are present throughout the manuscript.
Round 2
Reviewer 1 Report
Comments and Suggestions for Authors
It can be confirmed that the authors have revised the text and figures on basis of the review and addressed the points that were raised. However, there remains uncertainty about the significance of the reported findings/hits. Based on looking at available knockdown effects of the hits BRSK1, STK32C and STK40 on DepMap (see attached pdf), none stand out as potent "hits" in the models with the strongest known Hippo pathway aberration; NF2 mutation (nor the used model cell lines MDAMB231 and 468). Should a target be important in mediating molecular processes regulating YAP phosphorylation/dephosphorylation, and thus YAP/TEAD mediated signaling, NF2 mutant cells should be vulnerable to inhibition of such target(s). Moreover, the reviewer´s group has performed a near identical kinome-wide siRNA screen to look for nuclear localization of YAP (with antibody based detection) in the NF2 mutant mesothelioma cell line NCIH2052 (which is one of the tool model of all companies working on TEAD inhibitors currently), and in this comparison only the known "hits" are overlapping while none of the three reported hits BRSK1, STK32C and STK40 score as hits reducing nuclear YAP in these cells (see attached pdf). (however many other interesting hits do overlap suggesting that maybe more detailed re-assessment of the screen results should be performed. Maybe there are more widely valid YAP/TAZ signaling associated hits in the screen beyond BRSK1, STK32C and STK40 ). Last, the fact that BRSK1, STK32C and STK40 don´t seem to validate in e.g., DepMap, could point to certain breast cancer cell selective biological pathways/processes to which these genes contribute, which are not active in the cancer types which show more potent dependency on YAP/TAZ. (breast cancer is largely seen as a not suitable indication for TEAD inhibitors in the clinic. This is evident also by the fact that none of the current described TEAD inhibitors are being tested in breast cancer patients. Pathway seems non-essential in this context). That said, I think instead of you now publishing the names of these three genes together with the screen results, you could combine the screens with a comprehensive in silico evaluation of the "hits", and then validate and nominate some additional/other hits that show higher degree of validity in true YAP/TAZ dependent models. (though MDAMB231 has a NF2 mutation, its not dependent on YAP)

Reviewer 2 Report
Comments and Suggestions for Authors
The authors have adequately revised their manuscript. I recommend publication of the manuscript.
Author Response
Reviewer did not provide additional comments.